# A committed fourfold increase in ocean oxygen loss

Andreas Oschlies [1,2] ✉

Less than a quarter of ocean deoxygenation that will ultimately be caused by historical $CO_2$ emissions is already realized, according to millennial-scale model simulations that assume zero $CO_2$ emissions from year 2021 onwards. About 80% of the committed oxygen loss occurs below 2000 m depth, where a more sluggish overturning circulation will increase water residence times and accumulation of respiratory oxygen demand. According to the model results, the deep ocean will thereby lose more than 10% of its pre-industrial oxygen content even if $CO_2$ emissions and thus global warming were stopped today. In the surface layer, however, the ongoing deoxygenation will largely stop once $CO_2$ emissions are stopped. Accounting for the joint effects of committed oxygen loss and ocean warming, metabolic viability representative for marine animals declines by up to 25% over large regions of the deep ocean, posing an unavoidable escalation of anthropogenic pressure on deep-ocean ecosystems.

[1] GEOMAR Helmholtz Centre for Ocean Research Kiel, 24105 Kiel, Germany. [2] Kiel University, 24098 Kiel, Germany. ✉email: aoschlies@geomar.de

The Earth system is not in equilibrium with current levels of atmospheric $CO_2$ that have experienced an increasingly rapid growth as a result of anthropogenic $CO_2$ emissions, with about half of all anthropogenic $CO_2$ emitted during the past 35 years only[1]. Theory and models nevertheless predict that global-mean surface temperatures would essentially stop rising further and remain relatively stable for many decades to centuries once $CO_2$ emissions are stopped[2–5]. This is also implicit in the concept of transient climate response to cumulative $CO_2$ emissions (TCRE[6,7]), which provides the scientific rationale for relating temperature targets to remaining carbon budgets. Radiative forcing from non-$CO_2$ greenhouse gases and aerosols can lead to some variations in the relation between cumulative $CO_2$ emissions and warming[8,9]; however, given their shorter atmospheric lifetimes, temperature stabilization appears possible even for non-zero emissions of these substances[8]. As the explicit call for balancing sources and sinks of greenhouse gases in the Paris Agreement in 2015[10], the goal of achieving net zero $CO_2$ emissions to stabilize global-mean surface temperatures has gained substantial traction in climate politics and scenario development. Accomplishing this goal must, however, not be regarded as automatically stopping the increase in climate damages.

Even though global-mean surface temperatures are expected to remain stable when $CO_2$ emissions are stopped, many components of the Earth system will continue to respond to the anthropogenic perturbation with their inherent response timescales and inertia[2,11–16], producing committed impacts long after emissions are stopped. The uptake of $CO_2$ by the ocean essentially gives rise to a slow multi-centennial decline of atmospheric $CO_2$ and the associated radiative forcing. According to current models, the remaining radiative forcing during this phase of ocean adjustment to the anthropogenic perturbation is closely balanced by oceanic heat uptake[3,4]. The relative stability of global-mean surface temperatures within a few tens of degrees upon ending $CO_2$ emissions[5] thus comes at the expense of increasing acidification and ocean warming[11]. Ocean warming and associated thermal expansion of seawater adds to committed sea-level rise[12,13], which primarily results from committed melting of inland ice[14,15]. Committed changes in terrestrial ecosystems have also been reported[16]. This study investigates committed changes in marine oxygen levels that are already declining at a previously unexpected pace in response to anthropogenic $CO_2$ emissions[17,18], and that act, together with warming and acidification, as key stressors on marine ecosystems[19]. Such committed changes will also have to be accounted for when assessing ecological and socio-economic impacts caused by anthropogenic $CO_2$ emissions[20], even if they materialize well after emissions are stopped and global-mean surface air temperatures have stabilized.

## Results

**Committed change in oceanic oxygen inventory**. An emission-driven numerical Earth system model of intermediate complexity[21], which is calibrated to simulate observed climate properties and oxygen distributions[22,23], is employed to examine what would happen if $CO_2$ emissions were stopped by the end of year 2020. Forced with historical $CO_2$ emissions until 2010 and emissions corresponding to the Reference Concentration Pathway (RCP) 8.5 high-emission scenario[24] until year 2020 and closely agreeing with the actual emission data available until now[1], the model simulates atmospheric $CO_2$ levels of 411 µatm and global annual-mean surface air temperatures 1.03 °C above pre-industrial in year 2020, in agreement with observations. The simulated ocean heat uptake is consistent with the recent observational estimates of $1.29 \pm 0.79 \times 10^{22}$ J yr$^{-1}$ for the entire water column over the period

1991–2016[25], for which the model yields $1.15 \times 10^{22}$ J yr$^{-1}$ and of $33.5 \pm 7.9 \times 10^{22}$ J for the upper 2000 m over the period 1960–2015[26], for which the model yields $31.6 \times 10^{22}$ J. The simulated oceanic carbon uptake of 29.7 Pg C for the period 1994–2007 also agrees well with the recent observational estimate of $29 \pm 5$ Pg C[27]. After emissions are stopped by the end of 2020, simulated surface air temperatures increase by another 0.04 °C within 7 years before slowly leveling off at about 0.01 °C above the year 2020 temperatures towards the end of the twenty-first century and staying within ±0.02 °C at this level until year 2650 (Fig. 1a).

A major deep-convection event in the Southern Ocean after year 2650 sets off a substantial release of heat and $CO_2$ from the ocean to the atmosphere and surface air temperatures rise abruptly by 0.4 °C within 50 years. Such deep-convection events have been found in a number of climate models under prescribed elevated atmospheric $CO_2$[28,29] and continued $CO_2$ emission pathways[30]. In the zero-emission commitment scenario investigated here, the combination of stabilized surface temperatures with warming subsurface waters of North Atlantic origin eventually renders stratification unstable in the Southern Ocean, triggering deep convective overturning after year 2650. This leads to enhanced ventilation of the deep ocean from the south and an eventual re-oxygenation of the deep ocean also seen under continued emissions[29,31]. Simulated globally averaged ocean temperature increases from a pre-industrial value of 3.07 °C to 3.23 °C in year 2020 and shows continued warming to 3.70 °C

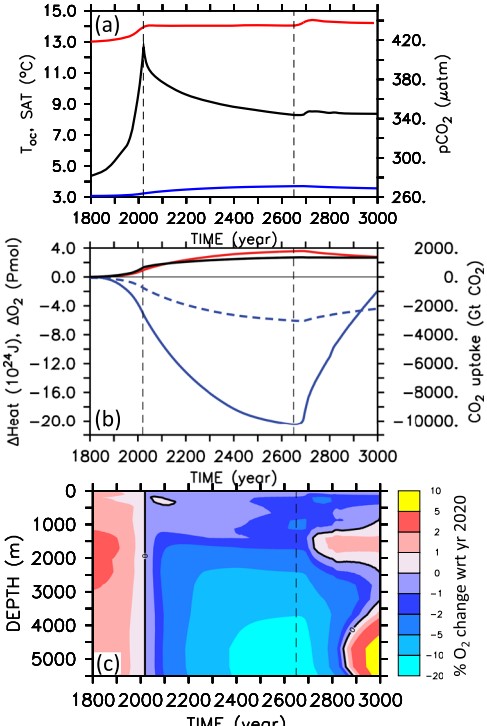

**Fig. 1 Simulated temporal evolution of ocean indicators. a** Simulated global annual-mean surface air temperature (SAT, red), atmospheric $pCO_2$ (black), and ocean mean temperature (Toc, blue). **b** Change in ocean inventory of heat (red), $CO_2$ (black), dissolved oxygen ($O_2$, blue), and its abiotic (solubility) component (abiot$O_2$, dashed blue) with respect to year 1800. **c** Change in laterally integrated oxygen concentration as a function of depth over time, expressed in percent change relative to year 2020. Vertical dashed black lines mark the end of year 2020 when emissions stop and the end of the evaluation period in year 2650, just before the onset of the Southern Ocean deep-convection event.

until year 2650, followed by a subsequent gradual cooling to 3.56 °C in year 3000 (Fig. 1a).

There is substantial uncertainty, requiring further study, about the relative roles in heat and freshwater forcings and the degree of realism in simulated Southern Ocean processes in current climate models[30]. Therefore, I here focus my analysis of committed changes on the time until year 2650, by which all model properties show an asymptotic adjustment to the stabilized surface temperatures. I regard the still poorly understood enigmatic deep-convection events in this and other models' Southern Ocean as a plausible but as-yet uncertain tipping element in the climate system, with the potential of decadal-scale global-mean temperature changes of a few tens of degrees. A more detailed investigation of this phenomenon will be presented elsewhere.

The stabilization of global-mean surface air temperatures after emissions stop by the end of year 2020 is a result of the close cancellation of the warming effect from declining ocean heat uptake and the cooling effect of declining atmospheric $CO_2$[3,5] (Fig. 1a, b). According to the model, the committed future uptake of $CO_2$ from the atmosphere (720 Gt $CO_2$ until 2650) is larger than all the $CO_2$ the ocean has taken up until 2020 (634 Gt $CO_2$), whereas the committed future ocean heat uptake until year 2650 ($2.7 \times 10^{24}$ J) is even three times as large as the heat taken up until year 2020 ($0.9 \times 10^{24}$), leading to another 16 cm of unavoidable thermosteric global sea-level rise. The simulated oceanic oxygen inventory declines by 5.0 Pmol $O_2$ (1.8%) from pre-industrial year 1770 to 2020. This is consistent with the decline found in other Earth system models[32], but somewhat slower than inferred from observations that suggest a $4.8 \pm 2.1$ Pmol decline between 1960 and 2010 alone[17]. The simulated oxygen decline continues for several hundred years after stopping emissions in 2020 and asymptotically reaches a total loss of 20.3 Pmol $O_2$ until year 2650 (Fig. 1b), corresponding to 7.4% of the pre-industrial oxygen inventory. Similar to the magnitude of the committed heat uptake, the committed oxygen loss is more than three times as large as the oxygen loss that has occurred until the emissions stop by the end of year 2020.

There is a close proportionality of global-ocean heat gain and oxygen loss until year 2650 (Supplementary Fig. 1) with a ratio of $5.7 \pm 0.9$ nmol $J^{-1}$, consistent with earlier observational estimates of thermocline oxygen losses[33]. The total oxygen loss is three to four times larger than the direct solubility effect of warming shown in the form of an abiotic oxygen tracer in Fig. 1b. Only the Southern Ocean deep-convection event after year 2650 breaks this proportionality by bringing large volumes of deep waters with a high share of accumulated respiratory oxygen deficit into contact with the atmosphere and thereby increasing the air–sea oxygen flux and, in consequence, the oceanic oxygen inventory (Fig. 1b).

**Patterns and processes**. Despite the tight correlation of global-ocean heat uptake and oxygen loss, regional patterns of warming and deoxygenation are very different. Until year 2020, zonally averaged ocean warming is largest in the mid- and low-latitude near-surface waters (Fig. 2c) with vertically averaged warming being most prominent along the western boundary of the North Atlantic (Fig. 2a), which is also found by observational estimates covering the recent decades[34]. As the ocean warms from the top, shallow coastal regions show a relatively large vertically averaged temperature increase in all ocean basins. Oxygen loss, on the other hand, is most pronounced below the surface mixed layer and down to several hundred meters depth in mid- to high-latitude regions in the Southern Ocean, North Atlantic, and North Pacific (Fig. 2b, d), in good agreement with other modeling studies[32,35]. Similar to many other models[32,35], there is even a slight oxygen gain in the tropical thermocline that is in conflict with observations and deemed attributable to the model's failure to correctly reproduce temporal changes in the wind field[23].

The committed ocean warming between years 2020 and 2650 by about half a degree Celsius is relatively homogeneous in the subsurface waters (Fig. 2g), but largest in the North Atlantic, where the overturning circulation ensures spreading of warmer waters throughout the basin within several decades to a few centuries (Fig. 2e). Surface waters in immediate contact with the stabilized surface air temperatures show very little changes in both temperatures and oxygen concentrations (Fig. 2g, h). There are, however, considerable regional differences in the magnitude of committed oxygen changes in the ocean interior. Committed oxygen loss is largest around Antarctica and in bottom and deep waters throughout the world ocean (Fig. 2f, h), where simulated oxygen concentrations decline by typically 30 mmol/m³. The Atlantic shows little changes and, in some regions, even an oxygen increase due to the slowly increasing ventilation via the formation of North Atlantic Deep Water and the strengthening meridional overturning circulation (contours in Fig. 2h).

The different patterns of committed warming and deoxygenation can be explained by the direct impact of biological respiration and water residence times on interior-ocean oxygen concentrations, but not on temperatures. Although the solubility-driven oxygen changes (Supplementary Fig. 2a, b) tightly follow the patterns of ocean warming, the more sluggish overturning circulation leads to increases in simulated bottom and deep-water ideal age by several hundred years by the year 2650 (Fig. 3). The export of organic matter out of the surface ocean increases by about 4% between year 2020 and 2650 (Supplementary Fig. 3). As remineralization rates increase with increasing temperatures, remineralization tends to move to slightly shallower depths under ocean warming. The combined effect of an increase in export production and a shoaling of remineralization is a little committed change in respiratory oxygen consumption in the deep ocean. Changes are typically <0.01 mmol $O_2$ m$^{-3}$ yr$^{-1}$ (Supplementary Fig. 4) and thus at least one order of magnitude smaller than the rates required to explain the realized changes in deep-ocean oxygen concentrations (Fig. 2h). According to these model results, the committed loss of marine oxygen is predominantly caused by changes in ocean physics.

Although ocean warming directly determines the solubility-driven oxygen loss, which, in the model, maintains roughly 30% share of total deoxygenation all the way to year 2650 (Fig. 1b), the majority of the committed oxygen loss is a result of the increase in water residence time (Fig. 3c, d) particularly in the deep Southern Ocean and Pacific Ocean (Table 1). These waters are relatively well oxygenated ($O_2 > 150$ mmol m$^{-3}$), whereas low-oxygen waters, sometimes called hypoxic ($O_2 < 70$ mmol m$^{-3}$) or suboxic ($O_2 < 5$ mmol m$^{-3}$), are generally found beneath the surface mixed layer but within the upper few hundred meters. Between years 2020 and 2650, the volume of hypoxic and suboxic waters shows only a relatively small and steady committed increases by 8% and 15%, respectively (Fig. 4b). This is small compared to the 150% increase seen in an extended multi-millennial RCP 8.5 emission scenario[31].

## Discussion

Results of the Earth system model of intermediate complexity used in the current study indicate that ocean deoxygenation will continue for several centuries, and that, in fact, the committed loss of marine oxygen is more than three times larger than the oxygen loss realized until now. A simulation with a full Earth system model with prescribed doubling of atmospheric $CO_2$ and consecutive constant atmosphere (i.e., implying residual positive

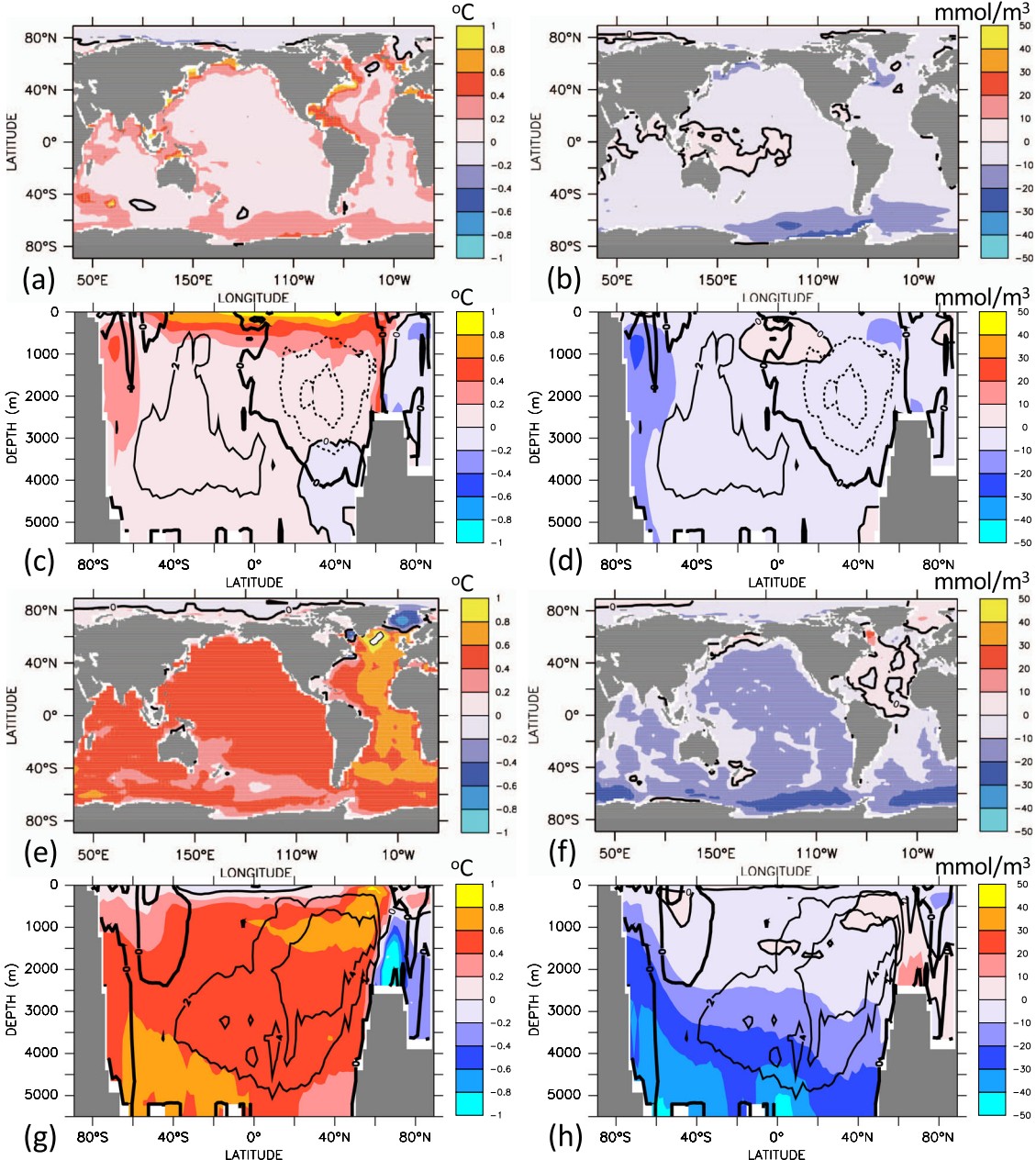

**Fig. 2 Simulated change in temperature (left column) and oxygen (right column). a**, **b** Vertically averaged difference year 2020 minus 1800; **c**, **d** zonally averaged year 2020 minus year 1800; **e**, **f** vertically averaged difference year 2650 minus 2020; and **g**, **h** zonally averaged difference year 2650 minus 2020. Contour lines indicate changes in the zonally averaged overturning stream function, with 2 Sv spacing of the iso-contours.

$CO_2$ emissions to maintain elevated atmospheric $CO_2$ levels) finds that the oceanic oxygen loss at the end of the increase in atmospheric $CO_2$ (3.6 Pmol) increases almost fivefold to 17.5 Pmol after another 650 years of constant atmospheric $CO_2$ levels[29]. Interestingly, that model also simulates a deep Southern Ocean convection event and subsequent onset of ventilation of the deep ocean from the south, providing some confidence in the robustness of the results reported here.

Simulated committed oxygen loss is largest in deep waters (Fig. 2h and Table 1) at oxygen concentrations relatively high compared to oxygen minimum zones typically located at a depth of a few hundred meters. Largest committed volumetric changes are simulated for waters with oxygen concentrations in the range between 230 and 270 mmol m$^{-3}$, which lose more than half of their volume between years 2020 and 2650, on the expense of

volume gains at lower oxygen classes between 200 and 230 mmol m$^{-3}$, and between 130 and 180 mmol m$^{-3}$ (Fig. 4). Although relatively little is known about deep-ocean ecosystems and their sensitivity to oxygen change, it is likely to be that animals have evolved to optimally exploit ambient oxygen levels[36]. Considering, at each location, species specifically adapted to local oxygen conditions, any loss of oxygen is expected to reduce habitats of individual species. The rate of oxygen decline is small, on average about 5 mmol m$^{-3}$ per 100 years until year 2650, but, according to the model and particularly in the deep ocean, unavoidable even if $CO_2$ emissions were stopped the end of year 2020. The generally low natural variability in the properties of deep water masses may make their ecosystems more vulnerable to oxygen changes than near-surface systems that are naturally exposed to large seasonal and interannual oxygen fluctuations. The speed of

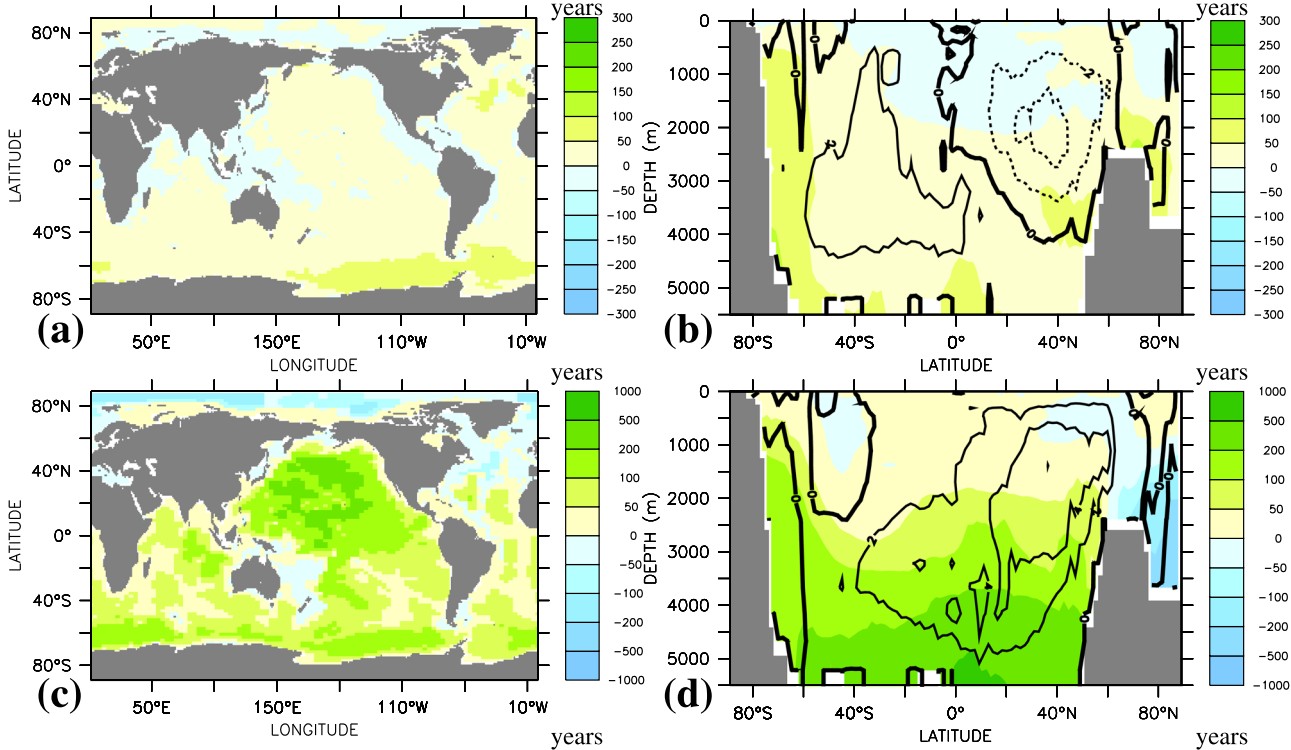

**Fig. 3 Changes in ideal age.** Simulated changes in vertically and zonally averaged water age for **a**, **b** year 2020 minus year 1800 and **c**, **d** year 2650 minus year 2020. Contours in **b** and **d** are changes in the overturning stream function (units Sv, with increments of 2 Sv) over the same time intervals.

**Table 1 Simulated changes in dissolved oxygen.**

| Oxygen change (Tmol) | 1770–2020 (0–1200 m) | 1770–2020 (>1200 m) | 2020–2650 (0–1200 m) | 2020–2650 (>1200 m) |
|---|---|---|---|---|
| Arctic (60N–90N) | −23.5 | −26.1 | −49.8 | 30.6 |
| North Atlantic (15N–60N) | −97.6 | −322.4 | −63.5 | 173.1 |
| eq. Atlantic (15S–15N) | 20.0 | −240.6 | −229.5 | −9.5 |
| South Atlantic (50S–15S) | −115.5 | −457.8 | −172.0 | −787.9 |
| North Pacific (15N–60N) | −215.5 | −192.2 | −19.6 | −2828.0 |
| eq. Pacific (15S–15N) | 138.6 | −291.5 | −136.7 | −3080.0 |
| South Pacific (50S–15S) | −149.0 | −508.6 | −70.2 | −2414.0 |
| eq. Indian Ocean (15S–15N) | 32.9 | −96.3 | −106.1 | −673.0 |
| S. Indian Ocean (50S–15S) | −111.5 | −360.4 | −39.0 | −1525.0 |
| Southern Ocean (90S–50S) | −605.7 | −1341.0 | −218.2 | −3042.0 |
| Global Ocean | −1126.0 | −3838.0 | −1112.0 | −14175.0 |

Changes are in Tmol for different regions and depth ranges from 0 to 1200 m and >1200 m, following ref. [17], and the two time period years 1770–2020 and 2020–2650.

model-derived anthropogenic warming relative to natural variability has recently been identified to be largest in mid and deep waters[37]. This would directly translate to the solubility-driven component of oxygen decline. Given that the solubility contributes less (<20%) than average (30%) to the total oxygen loss in deep waters (Supplementary Fig. 2c), the velocity of oxygen change can be expected to be even larger than that of warming.

A metabolic index[38] defined as the ratio of $O_2$ supply to the temperature-dependent resting $O_2$ demand of marine animals (until now sufficient information is available predominantly for upper ocean animals and it remains to be shown whether similar parameters can be confirmed experimentally for deep-sea animals[39,40]; see "Methods") and indicator of metabolically viable marine environments, declines by 10% to 25% over much of the deep ocean below 2000 m between year 2020 and 2650 (Fig. 5). Almost all of this reduction in the metabolic index results from oxygen decline. With species generally evolved such that their physiological capacity for oxygen supply matches the

maximum evolved demand at the available oxygen pressure[36,40], an oxygen decline of this magnitude can be expected to have substantial impacts on the still poorly explored deep-ocean fauna. This calls for more research efforts to explore the baseline of these systems before the unavoidable change will hit. A future recovery of oxygen concentrations as predicted by climate models after many hundred years of warming-induced deep-ocean deoxygenation[29,31], but until now based on a small set of model studies and requiring confirmation by further investigation, will likely come too late for many organisms. If RCP 8.5 $CO_2$ emissions continue until year 2100 and decrease linearly to zero in year 2300 as assumed in earlier studies with the same model[31], the marine oxygen loss will be even 2.5 times higher until year 2650.

On a positive note, present-day oxygen minimum zones in the upper few hundred meters experience, on average, only a little change once $CO_2$ emissions are stopped in the model (Figs. 2h and 4b). This indicates that rapid emission reduction can halt the

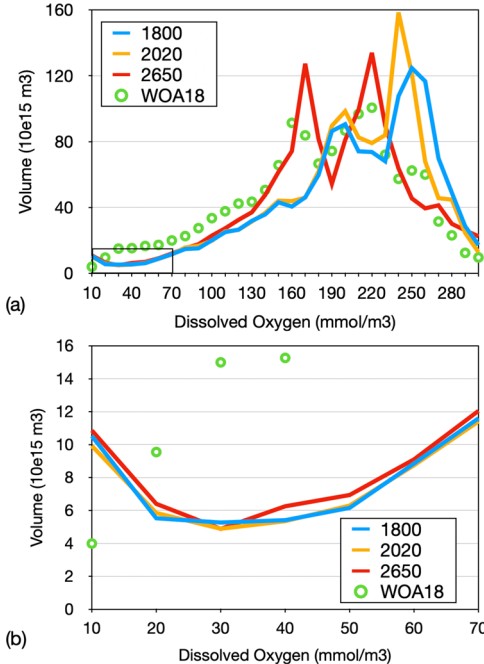

**Fig. 4 Simulated water volume ($10^{15}$ m$^3$) binned according to oxygen concentration with bin width of 10 mmol m$^{-3}$.** Blue refers to year 1800, orange to year 2020, and red to year 2650. Circles refer to the data of the World Ocean Atlas 2018[51]. **b** The enlarged low-oxygen range marked in the lower left corner of **a**.

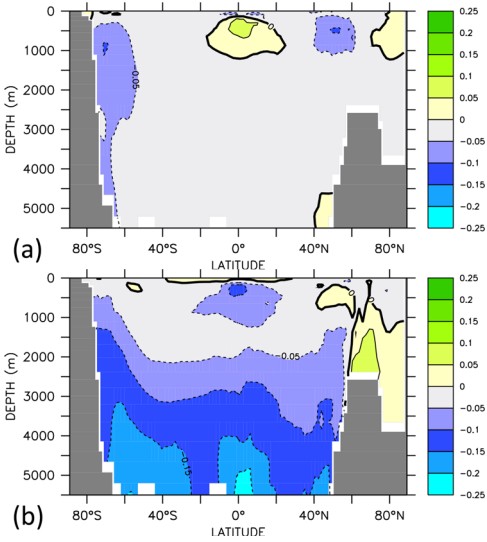

**Fig. 5 Zonally averaged relative changes in the metabolic index. a** Changes in metabolic index $\Phi$ (see "Methods") for year 2020 with respect to year 1800 and **b** for year 2650 with respect to year 2020.

increasing threat of deoxygenation in the upper ocean that holds much of the known biodiversity and provides the main ocean ecosystem services deemed societally relevant. For deep waters of the ocean, however, there is little hope that even an immediate stop of $CO_2$ emissions will avert a drastic decline in oxygen concentrations over the next decades to centuries, significantly reducing metabolically viable habitats of deep-sea animals. Although further analysis of models and data is required for a more robust and quantitative understanding of ocean deoxygenation, the deep ocean appears committed to turning into an as-yet unrecognized area where the slogan of the American Lung

Association "If you can't breathe, nothing else matters" will become reality for centuries to come.

## Methods

**Model configuration.** The University of Victoria Earth System Climate Model[21], version 2.8, is employed in the configuration described in ref. [31]. The ocean component is a fully three-dimensional primitive-equation model with 19 levels in the vertical ranging from 50 m near the surface to 500 m in the deep. It contains a simple marine ecosystem model including the two major nutrients, nitrate and phosphate, and two phytoplankton classes, nitrogen fixers and other phytoplankton, the former being limited by phosphate only. The micronutrient iron is not explicitly included in the model, which nevertheless achieves a reasonable fit to observed biogeochemical tracer distributions for the tuned biological parameters and mixing parameterizations[22]. For the assumed molar stoichiometry of $C : N : P : -O_2 = 112 : 16 : 1 : 169.6$, organic matter is degraded by aerobic remineralization ($-O_2 : PO_4 = 169.6$) as long as sufficient dissolved oxygen is available. In regions where oxygen concentrations fall below a threshold of 5 mmol $O_2$ m$^{-3}$, nitrate is used as an electron acceptor (denitrification, $-NO_3 : PO_4 = 119.68$, see ref. [41]). No other electron acceptors are simulated and remineralization stops whenever nitrate runs out, which does not happen in the model runs used here.

The ocean component is coupled to a single-level energy-moisture balance model of the atmosphere, a dynamic–thermodynamic sea ice component, and a terrestrial vegetation and carbon-cycle component. Continental ice sheets are prescribed and kept fixed in the present configuration. All model components use a common horizontal resolution of 1.8° latitude times 3.6° longitude. The current model version does not consider any fluxes across the water-sediment interface and also does not account for fluxes related to weathering on land. Oceanic phosphorus is thus strictly conserved. As the atmosphere contains about a hundred times as much oxygen as the ocean, any feedback of marine oxygen changes on atmospheric oxygen is neglected as in earlier studies (e.g., see ref. [31]).

**Model scenario.** The model is spun up for more than 10,000 years under preindustrial atmospheric $CO_2$ and is forced with historical $CO_2$ emissions from year 1765 until 2010 and emissions corresponding to the RCP 8.5 high-emission scenario[24] until year 2020. Solar forcing follows astronomical parameters and the atmospheric composition is assumed constant, except for $CO_2$. In the idealized model scenario aimed at studying the zero emissions commitment[42], $CO_2$ emissions are abruptly stopped at the end of year 2020 and kept zero thereafter until the end of the model run in year 3000. In model simulations that also stop emissions of non-$CO_2$ greenhouse gases and aerosols, the climate warms for a few years in response to the negative radiative forcing associated with the rapid decline of short-lived atmospheric aerosols, before a more gradual cooling sets in due to the decline in non-$CO_2$ greenhouse gases. After about a century, the response in such models is largely dominated by the long-lived $CO_2$ and global-mean temperatures converge to those obtained under elimination of $CO_2$ emissions alone[43]. Similar configurations of this model have been employed in earlier investigations of committed warming[3] and committed sea-level rise[44].

In the configuration employed here, eutrophication via runoff or atmospheric deposition of nutrients is not considered and ocean deoxygenation is thus entirely caused by effects related to anthropogenic $CO_2$ emissions. Anthropogenic effects of atmospheric nitrogen deposition have been studied elsewhere and have been found to be small compared to the ongoing warming-driven deoxygenation (e.g., see refs. [45,46]) as a result of stabilizing feedbacks in the nitrogen cycle[47].

**Metabolic index.** The metabolic index $\Phi$[38] is defined as the ratio of $O_2$ supply to the temperature-dependent resting $O_2$ demand of an organism and it combines temperature and p$O_2$ as indicators of metabolically viable marine environments. Here, not $\Phi$ itself but only changes of $\Phi$ are considered[48], employing the scripts provided by ref. [49]:

$$\frac{\Delta\Phi(t, t_0)}{\Phi(t_0)} = \frac{pO_2(t)}{pO_2(t_0)} \exp\left(\frac{E_0}{k_B}\left[\frac{1}{T(t)} - \frac{1}{T(t_0)}\right]\right) - 1 \qquad (1)$$

where $E_0$ describes the effect of temperature on the critical p$O_2$, i.e., the threshold oxygen partial pressure required for maintaining the resting metabolic rate (hypoxia vulnerability in ref. [40]) to temperature, which is different for different species. Parameter $k_B$ is the Boltzmann constant and p$O_2$ is the partial pressure of oxygen.

For $E_0$, the species average of ref. [40] is taken ($E_0 = 0.4$ eV). $E_0$ varies considerably among species[40] and may be different for deep-sea species, for which such information is not yet available. As an extreme example, a small negative value of $E_0 = -0.2$ eV, indicating an increase in the critical p$O_2$ with decreasing temperature, was found for a species living at the lower oxycline of the oxygen minimum zone in the eastern tropical Pacific[50] (where p$O_2$ tends to increase downwards with decreasing temperature). As changes in $\Phi$ are dominated by changes in p$O_2$ rather than temperature, the results are relatively similar even for a very low value of $E_0 = -0.2$ eV (Supplementary Fig. 5).

## Data availability

The data presented in the paper are from model simulations described in the Methods and are available at https://hdl.handle.net/20.500.12085/1a2adccc-bb63-4e38-ba57-43852fb2c2fc.

## Code availability

All model and analysis code is available at https://hdl.handle.net/20.500.12085/1a2adccc-bb63-4e38-ba57-43852fb2c2fc.

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

## Acknowledgements

I thank the members of the Global Ocean Oxygen Network (GO2NE) and the Marine Biogeochemical Research Unit at GEOMAR for fruitful discussions, in particular Iris Kriest and Wolfgang Koeve for essential help with the ferret analysis software and the presentation of the results. This is a contribution to the EU H2020 COMFORT project

that received funding from the European Union's Horizon 2020 research and innovation program under grant agreement number 820989 (project COMFORT, Our common future ocean in the Earth system–quantifying coupled cycles of carbon, oxygen, and nutrients for determining and achieving safe operating spaces with respect to tipping points). The work reflects only the author's/authors' view; the European Commission and their executive agency are not responsible for any use that may be made of the information the work contains.

## Author contributions

AO designed and carried out the model runs and wrote the paper.

## Funding

## Competing interests

The author declares no competing interests.
