## [Peer Review File · Nature Communications]

REVIEWERS' COMMENTS

Reviewer #1 (Remarks to the Author):

Thank you for the overall revision of the manuscript, and the clear reply to all provided comments. Most of my concerns have been replied to, and below I further provide some minor comments to support the draft improvement. I also suggest to provide research recommendations in the conclusion part of this paper.

Comments:

L.36-38: I am still puzzled with this sentence, as it does not reflect the still large unknown on aerosol forcing, and its impacts. I agree, and as also stated in the sentence – albeit not specified – on the timescales, but nevertheless does not capture the whole complexity.

L42-43: I think I understand the intend of this sentence, but – as it stands – is a matter of interpretation, and appears to be pessimistic. It is true that 'loss and damage' has not been specifically mentioned in the agreement, but one should not underestimate the challenge to achieve agreement at international policy level. A change of this sentence could be for example say 'Accomplishing this goal also needs to consider stopping the increase in climate damages.' or equivalent.

L.53-55: This is a matter of timescales, and should be precised. See also Palmer & McNeal (2014), doi:10.1088/1748-9326/9/3/034016. Moreover change 'oceanic uptake' to 'ocean heat storage' would be also more correct.

L.57: change 'warming of the deep ocean' to 'ocean warming'. The majority of ocean warming takes place in the upper 2000m depths, and however, the deep ocean warming is also important. I just think that the way it is written is misleading as it is not the deep ocean only.

L57-58: Also here, I recommend slight revision –the process of vertical heat redistribution is one part of the 'ocean warming complexity', and I recommend to see this as a whole, and thus recommend to stay with 'ocean warming' or equivalent.

L. 65: recommend to add a reference on the socioeconomic impacts from climate change

General on introduction: the introduction lacks to provide insight on the attribution of ocean deoxygenation. (eg AR5: decrease can be attributed in part to human influences with medium confidence...)

L.117-118: see also: <https://doi.org/10.5194/bg-17-2987-2020>

L.207: non-scientific language, and revision is needed on 'more quickly'.

L.208-209: 'oxygen loss realized until now' – consider change

General on conclusion: An outlook on recommendations for future analyses, a critical comment on gaps of this study which could be filled through future studies, and which are the new science pathways opened through the analysis / outcome of this study, and where science should further advance to fill these gaps is lacking, and are fundamental to provide to the community for further advance in climate science – these types of recommendations are essential.

Minor:

L. 26: Change to Earth system

L. 61: change to 'This study...'

Reviewer #2 (Remarks to the Author):

This is a second review of the manuscript "a committed fourfold increase in ocean oxygen loss" by A. Oschlies. As I stated earlier in the previous review, this study is timely and addresses important question about the future evolution of ocean deoxygenation using an intermediate complexity earth system model.

I believe that the author addressed both of my major concerns regarding the model dependence and unknown traits of deep water species. The response letter as well as the revision in the main text appear to be adequate responses, and I appreciate the added nuances by the revision to address these uncertainties. As far as my comments are concerned, I believe that the author address them and I support the publication of this manuscript. I would defer to other reviewers about the responses to other comments, but it appears that the author made adequate responses from my point of view.

Reviewer #3 (Remarks to the Author):

The author has satisfactorily addressed my concerns in the previous submission to Nature. I have only one minor comment.

1. The author refers to a complete lack of physiological data on deep-sea animals. Its true there are not many....but there are some. Temperature sensitivity of metabolic rate and Pcrit has been addressed in a few species in papers by Jim Childress, Jose Torres, David Cowles and others. A few of these were included in Deutsch et al., 2020. There were a few for which the temperature sensitivity of metabolism and Pcrit were very near zero (E values) and others which showed much more marked temperature sensitivity. One example reference is below.

THE RESPIRATORY RATES OF MIDWATER
CRUSTACEANS AS A FUNCTION OF DEPTH OF
OCCURRENCE AND RELATION TO THE OXYGEN
MINIMUM LAYER OFF SOUTHERN CALIFORNIA
JAMES J. CHILDRESS 1975.

Dear Reviewers,

Thank you very much for your very careful reading and constructive comments and suggestions for improvement of the manuscript! Unfortunately, the journal does not allow to acknowledge this in the acknowledgement section of the manuscript.

In the following I address all of the points raised by the reviewers, with answers to your comments provide in blue font color. Line numbers in my responses refer to the new version WITH track changes (line numbers differ from the version WITHOUT track changes).

In a final check of the manuscript I unfortunately realized that panel (a) of Figure 5S had contained the wrong parameter $E_0 = 0.4$ eV instead of $E_0 = -0.2$ eV. The correct figure is shown in the revised manuscript and shows a slightly more positive metabolic index, but does not change the message of figure or manuscript.

Referee #1 (Remarks to the Author):

Thank you for the overall revision of the manuscript, and the clear reply to all provided comments. Most of my concerns have been replied to, and below I further provide some minor comments to support the draft improvement. I also suggest to provide research recommendations in the conclusion part of this paper.

Thank you for your very constructive and detailed suggestions for further improvement of the manuscript!

Comments:

L.36-38: I am still puzzled with this sentence, as it does not reflect the still large unknown on aerosol forcing, and its impacts. I agree, and as also stated in the sentence – albeit not specified – on the timescales, but nevertheless does not capture the whole complexity.

Agreed. I have changed the sentence accordingly to ‘Radiative forcing from non-CO₂ greenhouse gases and aerosols can lead to some variations in the relation between cumulative CO₂ emissions and warming^{8,9}, but given their shorter atmospheric lifetimes, temperature stabilization appears possible even for non-zero emissions of these substances⁸.’ The reference to Matthews and Zickfeld (Climate response to zeroed emissions of greenhouse gases and aerosols. *Nature Climate Change*, 2, 338–341, 2012) has been added to refer to the complexity issue.

L42-43: I think I understand the intend of this sentence, but – as it stands – is a matter of interpretation, and appears to be pessimistic. It is true that ‘loss and damage’ has not been specifically mentioned in the agreement, but one should not underestimate the challenge to achieve agreement at international policy level. A change of this sentence could be for example say ‘Accomplishing this goal also needs to consider stopping the increase in climate damages.’ or equivalent.

I think I now understand your point and agree. I’ve changed the sentence to “...not be regarded as automatically stopping the increase in climate damages.”

L.53-55: This is a matter of timescales, and should be precised. See also Palmer & McNeal (2014), doi:10.1088/1748-9326/9/3/034016. Moreover change ‘oceanic uptake’ to ‘ocean heat storage’ would be also more correct.

I agree with the reviewer that this is a matter of timescales, but do not think that a text change is required. The timescale had already been mentioned in the preceding sentence (“...gives rise to a slow multi-centennial decline...”), and referred to in the sentence under consideration via

“...during this phase...”, making clear that the multi-centennial timescale is meant. This is in agreement with the finding of Palmer & McNeall for decadal and longer timescales.

L.57: change ‘warming of the deep ocean’ to ‘ocean warming’. The majority of ocean warming takes place in the upper 2000m depths, and however, the deep ocean warming is also important. I just think that the way it is written is misleading as it is not the deep ocean only.

Agreed. Thank you.

L57-58: Also here, I recommend slight revision –the process of vertical heat redistribution is one part of the ‘ocean warming complexity’, and I recommend to see this as a whole, and thus recommend to stay with ‘ocean warming’ or equivalent.

Agreed.

L. 65: recommend to add a reference on the socioeconomic impacts from climate change

Good point. I have added a new reference: Nordhaus, W. D., Revisiting the social cost of carbon, PNAS, 114(7), 1518, 2017.

General on introduction: the introduction lacks to provide insight on the attribution of ocean deoxygenation. (eg AR5: decrease can be attributed in part to human influences with medium confidence...)

Thank you. This information was indeed lacking and is not included (‘in response to anthropogenic CO₂ emissions’, line 60.

L.117-118: see also: <https://doi.org/10.5194/bg-17-2987-2020>

Thanks, now also cited (ref.5).

L.207: non-scientific language, and revision is needed on ‘more quickly’.

Changed to ‘will continue for several centuries’.

L.208-209: ‘oxygen loss realized until now’ – consider change

I find this phrase clear and to the point, and prefer not to change it.

General on conclusion: An outlook on recommendations for future analyses, a critical comment on gaps of this study which could be filled through future studies, and which are the new science pathways opened through the analysis / outcome of this study, and where science should further advance to fill these gaps is lacking, and are fundamental to provide to the community for further advance in climate science – these types of recommendations are essential.

Thanks for this suggestion. I made the call for more research efforts to explore the deep ocean ecosystems and their oxygen tolerance more explicit (line 254), as well as call for further investigation of the recovery of the marine oxygen inventory after many hundred years of warming (lines 257-258), I conclude with the request that ‘further analysis of models and data is required for a more robust and quantitative understanding of ocean deoxygenation’ (line 273/274).

Minor:

L. 26: Change to Earth system

done

L. 61: change to 'This study...'

done, plus adjusted language in preceding sentence.

Referee #2 (Remarks to the Author):

Reviewer #2 (Remarks to the Author):

This is a second review of the manuscript "a committed fourfold increase in ocean oxygen loss" by A. Oschlies. As I stated earlier in the previous review, this study is timely and addresses important question about the future evolution of ocean deoxygenation using an intermediate complexity earth system model.

I believe that the author addressed both of my major concerns regarding the model dependence and unknown traits of deep water species. The response letter as well as the revision in the main text appear to be adequate responses, and I appreciate the added nuances by the revision to address these uncertainties. As far as my comments are concerned, I believe that the author address them and I support the publication of this manuscript. I would defer to other reviewers about the responses to other comments, but it appears that the author made adequate responses from my point of view.

Thank you for your effort, earlier constructive comments and appreciation!

Referee #3 (Remarks to the Author):

The author has satisfactorily addressed my concerns in the previous submission to Nature. I have only one minor comment.

1. The author refers to a complete lack of physiological data on deep-sea animals. Its true there are not many....but there are some. Temperature sensitivity of metabolic rate and Pcrit has been addressed in a few species in papers by Jim Childress, Jose Torres, David Cowles and others. A few of these were included in Deutsch et al., 2020. There were a few for which the temperature sensitivity of metabolism and Pcrit were very near zero (E values) and others which showed much more marked temperature sensitivity. One example reference is below.

THE RESPIRATORY RATES OF MIDWATER
CRUSTACEANS AS A FUNCTION OF DEPTH OF
OCCURRENCE AND RELATION TO THE OXYGEN
MINIMUM LAYER OFF SOUTHERN CALIFORNIA
JAMES J. CHILDRESS 1975.

Thank you for this valuable input. I have now formulated this sentence more carefully (replaced 'only' by 'predominantly' and included the reference (line 246/247).